# Evaluation of Unmanned-Aerial-Vehicle-Integrated Control System Efficiency on the Basis of Generalized Multiplicative Criterion

Viktor Vititin, Maksim Kalyagin * and Valentin Kolesnichenko

Moscow Aviation Institute, Moscow 125993, Russia; vfvititin@mail.ru (V.V.); vekolesnichenko@mail.ru (V.K.)
* Correspondence: mukalyagin@mail.ru

**Abstract:** An unmanned aerial vehicle (UAV)-integrated control system is a set of functionally independent subsystems of the ground and space segments interacting with each other under the conditions of the stochastic nature of the external environment. There is an approach to evaluating its effectiveness based on a generalized multiplicative criterion, which takes into account the features of this system to the maximum extent. It is proposed to single out two particular criteria that characterize the reliability of a UAV and the effectiveness of the control system in relation to it. At the same time, the generalized criterion is a multiplicative convolution based not on the triangular-norm (*t*-norm) of the particular criterion, but of its correspondence functions, which in a certain way reflect its significance. It is shown that in the particular case of linear dependence of the correspondence functions, the generalized criterion coincides with the classical multiplicative convolution in the form of product of event probabilities. The proposed approach with minimal changes can be adapted to assess the effectiveness of data management systems in heterogeneous networks, process control systems, projects, logistics, etc.

**Keywords:** unmanned aerial vehicle (UAV); unmanned aircraft system traffic management (UTM); complex technical system (CTS); efficiency criterion; multiplicative convolution; triangular norm (*t*-norm); Hamacher *t*-norms

## 1. Introduction

One of the dynamically developing areas of robotics is unmanned aerial technologies, while UAVs are used in various fields of human activity [1,2]. The expansion of the range of areas of application of UAVs, their number, nomenclature and specialization necessitate the creation of UTM. Examples of such systems are the American Unmanned Aircraft System Traffic Management (UTM), the Chinese Aviation Operation Management System (UOMS) and the European U-Space [3–5]. These and other promising UAV control systems, which have a distributed structure and contain ground and space segments, should provide the following:

- Safety of UAV flights on the scale of one or several countries, including in the area of civil aviation flights;
- The possibility of solving various functional tasks within a single system;
- The possibility of using a UAV of various types and manufacturers to solve these problems.

From the point of view of system analysis, UAV control systems are CTSs and represent a set of functionally independent subsystems interacting with each other under the conditions of the stochastic nature of the external environment, and they are designed to achieve a common goal [6,7]. The most universal characteristic of a CTS is the efficiency of its functioning; however, in the scientific and technical literature, there is no single interpretation of this term and, as a rule, CTS efficiency is understood as follows [8–11]:

- Degree of compliance of the system with the purpose and adaptability to achieve the goals set during its creation;
- Possibility of achieving maximum results at given costs for the creation of the system.

The first definition assumes that the evaluation of the effectiveness of a CTS is based on the achievement of the targets for which it is developed, and the second one reflects its economic feasibility, which is not the subject of this article. In this article, we will use the first definition, while we assume that the quality of solving the target problems facing a complex system is determined by the result of achieving specific goals. In other words, through the effectiveness of a CTS, we will understand the property of the system to fulfill the set goal under given conditions and with a certain quality. At the same time, the performance indicators of a CTS should characterize the degree of adaptability of the system to solve the tasks assigned to it and are generalizing indicators, which in turn depend on local indicators (reliability, safety, etc.).

In turn, a quantitative measure of quality of solving target problems is the magnitude of the criterion for the effectiveness of solving functional problems assigned to a CTS. There can be quite a lot of such criteria, while they, as a rule, are stochastic and represent probabilistic quantities; for example, reliability is the probability that the system/subsystem will be operational for a certain time, efficiency is the probability that the system/subsystem response time will not exceed a given value, etc. Therefore, they are often called partial criteria or performance indicators (we will use the first term).

Thus, the assessment of the effectiveness of a CTS as a whole should be carried out taking into account the global criterion which is a vector quantity and in a certain way integrates a particular criterion. This problem belongs to the class of multicriterial problems [12–14] and the most common way to solve it in practice is convolution, i.e., transition from a vector function to a scalar one. Such a transition from a multicriterial problem to a single-criterion one is carried out by combining all particular criteria of $f_i(x)$ into one global criterion of $f(x) = F(f_1(x), f_2(x), \ldots f_n(x))$. In this case, the solution of the problem is reduced to finding the extremum of the only function $f(x)$, and the form of the function is determined by how the contribution of each particular criterion can be represented.

There are various approaches to choosing the form of the function $f(x)$, and among which, the most popular are additive convolution and multiplicative convolution, when the global criterion is, respectively, the sum or product of all particular criteria. There are also various variations in this type of convolution by introducing the priorities of partial criterion, weight coefficients, using non-linear convolutions, etc. [14–16]. A detailed review of the scientific papers on this topic is given in [17]. One of the main disadvantages of the known methods of convolution in relation to the evaluation of the effectiveness of the UAV control system is the equivalence of the partial criterion, when low values of some partial criteria are compensated by high values of other partial criteria. This circumstance can lead to erroneous decisions when designing and upgrading the system, when an option that combines not the best particular criterion can be chosen as the best one.

The purpose of this work is to develop an approach to evaluate the effectiveness of a UAV flight control system using a generalized multiplicative criterion, which to a large extent eliminates the shortcomings noted above and takes into account the features of this system to the maximum extent. The essence of the approach is in the fact that two particular criteria are distinguished, whereby one of which characterizes the reliability of a particular type of UAV, and the second characterizes the effectiveness of the control system in relation to it. At the same time, the generalized criterion is formed not on the basis of the particular criterion, but on the basis of the correspondence functions, which in a certain way reflect their significance. Multiplicative convolution of the correspondence functions is carried out using the *t*-norm.

It should be noted that this article was written as part of a grant from the Russian Science Foundation for the development of control models in heterogeneous networks, and the proposed approach is applicable to almost any mobile subscriber, including aircraft,

ground, surface and underwater unmanned vehicles. In addition, with slight modernization, this approach can be proposed to assess the effectiveness of process control systems, logistics, projects, etc. At present, based on the proposed approach, there is a methodology for assessing the effectiveness of controlling a swarm of UAVs.

This article is devoted to evaluating the effectiveness of the control system for a private class of unmanned vehicles, UAVs, which have certain features. In accordance with this, the article has the following structure:

- A "Theoretical basis" section, which substantiates the use of the efficiency criterion of the UAV control system based on multiplicative convolution and the need to move from probabilities to correspondence functions;
- A "Methodology" section, which selects specific types of correspondence functions in terms of probability and reliability, taking into account the specifics of UAV use;
- A "Results" section, which justifies the choice of the *t*-norm as a convolution of the correspondence functions and provides some calculation results;
- A "Discussion" section, in which it is proposed that one uses the modified *t*-norm (Hamacher's *t*-norm), and it also substantiates the possibility of using the proposed approach to assess the effectiveness of other CTSs and the direction of further development of the methodology.

## 2. Theoretical Basis

The task of studying the effectiveness of a UAV control system is formulated as follows:

- There are a number of functional (target) tasks that this system must solve, for example, cargo delivery, monitoring of important infrastructure facilities, environmental monitoring, agriculture, etc.;
- Requirements for their achievement are set, for example, in the form of a condition that the probability of solving the target problem should not be less than a given value;
- For each of these tasks, there is a specific set of UAVs designed to solve it and having known characteristics, including reliability;
- It is necessary to determine one or more UAV types that best meet the specified requirements for solving a specific target task.

To solve this problem, there is the following approach. In the first stage, only one UAV is considered and its particular efficiency criterion is determined. Such an efficiency criterion should meet the following basic requirements [18,19]:

- To characterize the object as a whole;
- To provide the possibility of obtaining its quantitative assessment with the required reliability;
- To have clearly defined boundaries of the area of its change.

With regard to a UAV, the most informative particular criterion is its reliability, which is numerically characterized by the probability of failure-free operation [20,21].

The second stage involves the division of the process of solving the target task that the UAV must perform into certain stages. For example, if the target task is the delivery of goods, then such stages are takeoff, finding a given route to the recipient of the cargo, landing at the place of unloading, unloading itself, taking off, reaching a given route and landing at a given place. Each of these stages is controlled by the control system; however, all of them are subject to the influence of destabilizing environmental factors (wind, rain, birds, electromagnetic effects, artificial and natural obstacles, etc.). The influence of these factors leads to the fact that, at a certain stage, UAVs may fail as a result of their influence and the target task will not be fulfilled, i.e., there is a non-zero probability of system failure at a certain stage. Therefore, the probability of solving the target problem is a multiplicative convolution of the probabilities of successful completion of all stages of UAV flight.

Since the reliability of a UAV is a probabilistic value, and the probabilities of performing individual stages can be considered to be independent of each other, it is advisable to

use the following multiplicative convolution as a global criterion for the effectiveness of a UAV flight control system [18,19,22,23]:

$$W = p_g \prod_{i=1}^{n} p_i, \tag{1}$$

where $p_i$ is the probability of completing the $i$-th stage; $p_g$ is the UAV reliability; and $n$ is the number of steps that the UAV must complete.

The global criterion (1) has a simple structure and a clear physical meaning (the global criterion is the product of the particular criterion), but there are the following significant disadvantages:

- The criteria are indifferent to the probability distribution $p_i$, $\left(i = \overline{1, \ldots, n}\right)$ over the flight stages, but depend only on the value of their product. In other words, if we have some set of values $\{p_1, p_2, \ldots, p_n\}$, then with any distribution of this set over stages, the value of the criteria will not change;
- The criteria are indifferent to the relative changes in the probabilities $p_i$, $\left(i = \overline{1, \ldots, n}\right)$, i.e., the relative increase or decrease in any of them is compensated by exactly the same relative decrease or increase in any of the others.

We consider, in more detail, the disadvantages of a criterion of the form of (1) using the classical example of two particular criteria:

$$W = W(p_e, p_g) = p_e \cdot p_g, \tag{2}$$

where $p_e = \prod_{i=1}^{n} p_i$ is the probability that an absolutely reliable UAV will solve its target problem.

It follows from expression (2) that both variants of the UAV, having different probabilities $p_{ei}$ and $p_{gi}$ $(i = 1, 2)$, under the condition $p_{e1}p_{g2} = p_{e2}p_{g2}$, have the same values of criterion (2), i.e., they are equivalent. This decision has an error, which is confirmed by the following simple example. Let the first variant be characterized by the probabilities $p_{e1} = 0.5$, $p_{g1} = 0.9$, and the second be characterized by the probabilities $p_{e2} = 0.9$, $p_{g2} = 0.5$. The global criterion for these options has the same value, namely 0.45.

$$W_1(p_{e1}, p_{g1}) = W_2(p_{e2}, p_{g2}) = 0.45. \tag{3}$$

However, it is difficult to imagine that, according to (3), the second option, for which every second UAV is inoperable, can be chosen as the best of the two options. It follows from this that preference should be given to the first option, i.e., the equality of the values of the main criterion does not guarantee equivalence. However, from a purely mathematical point of view, the considered options are equivalent, because in the end they lead to the same value of the criteria $W(p_e, p_g)$, whereby the physical essence of which is the probability of solving the target problem.

To eliminate this disadvantage of multiplicative convolution of partial criteria, restrictions are usually introduced on their lower bound of values in the form of inequalities $p_i \geq p_{0i}$. However, the presence of restrictions in the form of inequalities can lead to the loss of a feasible solution and does not remove the internal inconsistency of the criterion in the form of (1), which is not able to clearly separate different options [18,19].

In view of this, an urgent task is such a modification of criterion (1), which allows for the elimination of the above disadvantages whilst retaining its structure, i.e., multiplicative convolution. One of the possible solutions is to use correspondence functions $s_i(p_i)$ instead of the partial criterion $p_i$, which would reflect the significance of the achieved value of the partial criterion. In this case, by the correspondence function, we mean a real one-place function, (one-place function) $s : [0, 1] \to [0, 1]$, which satisfies the following conditions: the function $s(p)$ must:

- Have a clear physical meaning;
- Be continuous and not have jumps and breaks;

- Be non-decreasing and take values at the ends of the domains $s(0) = 0$ and $s(1) = 1$.

## 3. Methodology

As was shown above, the global criterion for the effectiveness of the UAV control system can be represented as a multiplicative convolution of two partial criteria:

- UAV reliability, i.e., the probability that in the process of performing the target task it will be operational;
- Probability of achieving the goal (solving its target task) with an absolutely reliable UAV, which is a multiplicative convolution of the probabilities of completing each of the flight stages.

For each of these partial criteria, we define the type of correspondence function, while for simplicity of presentation we will call them the correspondence function in terms of reliability and the correspondence function in terms of probability.

According to this approach, the methodology is divided into the following steps:

1. Selection of specific types of probability correspondence function based on the analysis of existing criteria for evaluating the effectiveness of UAV control systems.
2. Choice of specific types of the matching function in terms of reliability, taking into account the fact that a UAV with reliability below the given one is not considered when comparing options.
3. Choice of a specific type of convolution for the selected correspondence functions.
4. Calculations.

### 3.1. Probability Matching Function

To build a probability matching function, we consider some criteria proposed in the scientific and technical literature in relation to the comparative evaluation of the effectiveness of a UAV control system [14,18,19]:

- $p$ is the probability of reaching the target with one UAV. When comparing options according to this criterion, preference is given to the option for which this probability has the highest value.
- $\omega = 1/p$ is the average number of UAVs needed to reach the goal. This criterion allows indirectly evaluating the possible material costs for solving the target problem.
- $n = \frac{\ln(1-W_s)}{\ln(1-p)}$ is the required number of UAVs to achieve the goal with a given probability $W_s$. This criterion relates the probability $p$ of reaching the goal with one UAV with the value $n$ of the required number of UAVs for a given probability of reaching the goal $W_s$.

As follows from the analysis of the criteria listed above, the third criterion bears the greatest information load, which allows for comparing the values $n$ of the required number of UAVs for different options for a given probability of achieving the goal $W_s$.

In this case, the ratio of the required number of UAVs for two options can be represented as follows:

$$\overline{n} = \frac{n_1}{n_2} = \frac{\ln(1-p_2)}{\ln(1-p_1)}. \tag{4}$$

The choice of specific types of probability correspondence function $s_p(p)$ is not a strictly formalized task; therefore, such a choice was made heuristically by analyzing various known functions.

$$s_p(p) = \begin{cases} \frac{\ln(1-p)}{\ln(1-W_s)}, & if\ p < W_s, \\ 1, & if\ W_s \le p \le 1, \end{cases} \tag{5}$$

or

$$s_p(p) = \begin{cases} \frac{\ln(1-p)}{\ln(1-W_s)}, & if\ 0 \le p \le p^*, \\ \end{cases} + \frac{\cdot\frac{p-p^*}{1-p^*},\ if\ p^* < p \le 1,}{} \tag{6}$$

where the values of $p^*$ and $W^*$ are determined by dependencies:

$$p^* = 1 - \exp[-2], W^* = \frac{\ln(1-p^*)}{\ln(1-W_s)}.$$

The function $s_p(p)$ in the form of (4) makes it possible to estimate the degree of compliance of the value $p$ with respect to the given efficiency $W_s$ in reciprocal values of the required number of means $n$.

For example, if the condition $p \geq W_s$ is satisfied for the considered application of the UAV, then it fully corresponds to the task, and for it, the probability matching function is $s_p = 1$. If $p < W_s$, then $< 1$, and the smaller the value of $p$, the smaller the value of the probability correspondence function.

The correspondence function in the form of (4) is continuous, but has an indifference segment (constant and equal to 1) on the interval $p \in [W_s, 1]$. The correspondence function in the form of (5) is continuous and has a continuous first derivative. Figure 1 shows the probability correspondence functions $s_p(p)$ for the above Formulas (4) and (5).

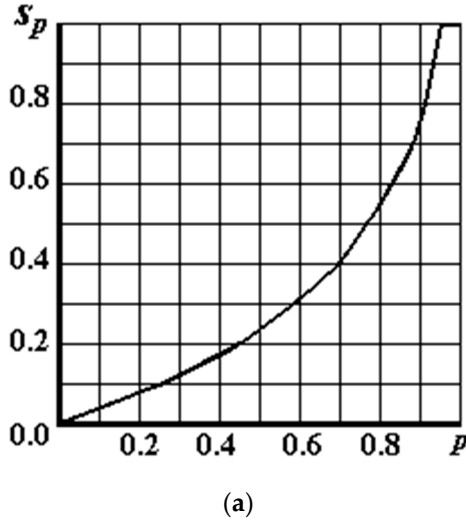
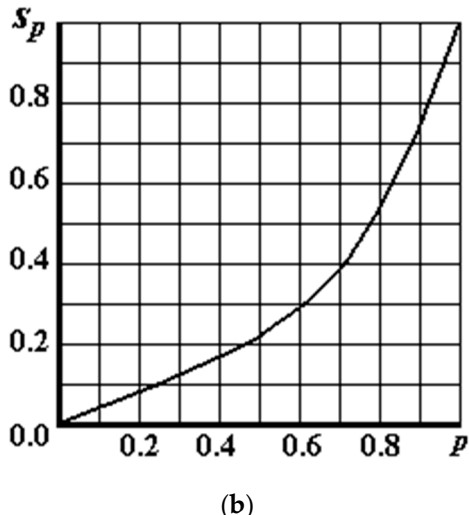

**(a)**                                    **(b)**

**Figure 1.** Probability matching functions according to (**a**) dependency (5) and (**b**) dependency (6).

### 3.2. Reliability Matching Function

The need to introduce the reliability matching function $s_g(p_g)$ is due to the fact that the quality of the UAV is not a linear function of the reliability $p_g$, which is implied when using the criterion in the form of (1). So, for example, if the reliability of the UAV is below some threshold $p_g^*$, then this UAV is not competitive and should not be considered at all when compared with other options.

This fact should be reflected in the following way: the reliability matching function with a value of $p_g$ below a certain threshold $p_g \leq p_g^*$ is equal to zero, i.e., $s_g(p_g \leq p_g^*) = 0$. It is also obvious that for the value $p_g = 1$, it is natural to take $s_g(p_g = 1) = 1$, i.e., the UAV fully complies with the requirements for reliability.

We formulated the requirements that the reliability matching function $s_g(p_g)$ must satisfy:

- The function $s_g(p_g)$ is defined on the interval $[0, 1]$ and its range of values also belongs to the interval $[0, 1]$;
- If the reliability of the UAV $p_g$ satisfies the inequality $p_g < p_g^*$, then the reliability matching function is $s_g(p_g \leq p_g^*) \approx 0$;
- The function $s_g(p_g)$ must be non-decreasing in the variable $p_g$ and for the value $p_g = 1$ it is natural to take $s_g(p_g = 1) = 1$;

- In the vicinity of the points $p_g = p_g^*$ and $p_g = 1$, the change in $s_g(p_g)$ should be smooth, i.e., have no jumps or breaks.

As in the case of the probability matching function, the choice of the $s_g(p_g)$ function was carried out heuristically, and as a result of which, we settled on the following option:

$$s_g(p_g) = \exp\left\{-\left|\frac{p_g - 1}{\delta}\right|^n\right\}. \tag{7}$$

This function has two degrees of freedom: the exponent $n$ allows changing the form of the function, and the denominator $\delta$ defines the width of the branch. Function (6), whereby the form of which is shown in Figure 2, satisfies all of the above requirements for the reliability matching function. In this case, the values of the coefficients $n$ and $\delta$ are determined by the method of expert assessments.

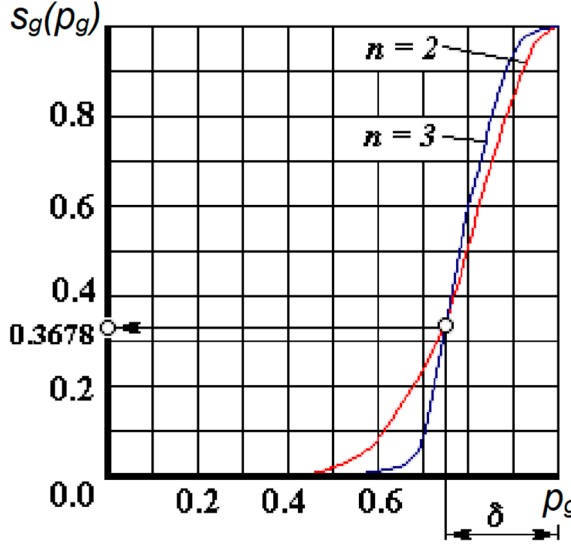

**Figure 2.** Reliability matching function.

## 4. Results

After the conformity functions have been obtained, it is necessary to bring them to the global criterion via convolution. This operation can be performed using the *t*-norm, which is the real two-place function $T : [0.1] \times [0.1] \to [0.1]$, and it satisfies the following conditions [19,24,25]:

- Limitations:

$$T(0.0) = 0; T(s(*), 1) = T(1, s(*)) = s(*);$$

- Monotony:

$$T(s_1(*), s_2(*)) \leq T(s_3(*), s_4(*)), \; if \; s_1(*) \leq s_3(*), s_2(*) \leq s_4(*);$$

- Commutativity:

$$T(s_1(*), s_2(*)) = T(s_2(*), s_1(*));$$

- Associativity:

$$T(s_1(*), T(s_2(*), s_3(*))) = T(T(s_1(*), s_2(*)), s_3(*)). \tag{8}$$

In the *t*-norm, the functions of $s_i(*)$ are unary real functions $s(*) : [0.1] \rightarrow [0.1]$, and the correspondence functions $s_p(p)$ and $s_g(p_g)$ obtained above were found according to the particular criteria of $p$ and $p_g$.

Therefore, the *t*-norm $T(s_p(p), s_g(p_g))$ determines the convolution of the correspondence functions $s_p(p), s_g(p_g)$, and ultimately the convolution of the particular criteria of $p$ and $p_\Gamma$, since $T(s_p(p), s_g(p_g)) = T(p, p_g)$.

We denote the function $T(p, p_g)$ of two partial criteria through the previously used notation $W_{np}(p, p_g)$, i.e., the set $W_{np}(p, p_g) = T(p, p_g)$, and use the reduced efficiency function to distinguish it from the efficiency function $W = p \cdot p_g$, given by relation (2).

In the scientific and technical literature, various types of *t*-norms are given, whereby the simplest of them are the following [19,24,25]:

- $T(s_p(p), s_g(p_g)) = \min[s_p, s_g]$;
- $T(s_p(p), s_g(p_g)) = s_p \cdot s_g$;
- $T(s_p(p), s_g(p_g)) = \max[0, s_p + s_g - 1]$.

In further studies, we will use the *t*-norm most appropriate to the problem under consideration in the following form:

$$T(s_p(p), s_g(p_g)) = W_{np}(s_p(p), s_g(p_g)) = s_p(p) \cdot s_g(p_g).$$

Then, the expression for the reduced convolution criterion can be written as follows:

$$W_{np}(s_p(p), s_g(p_g)) = \begin{cases} \frac{\ln(1-p)}{\ln(1-W_s)} \exp\left\{-\left|\frac{p_g-1}{\delta}\right|^n\right\}, & if\ p < W_s, \\ \exp\left\{-\left|\frac{p_g-1}{\delta}\right|^n\right\}, & if\ W_s \le p \le 1. \end{cases} \tag{9}$$

Due to the associativity property for the *t*-norm in the case of $n$ particular criterion $p_i, i = \overline{1, n}$, we find the following:

$$W_{np} = \prod_{i=1}^{n} s_i(p_i). \tag{10}$$

As follows from the comparison of dependencies (1) and (9), they have the same structure, but differ only in that dependence (1) folds the particular criterion, and dependence (9) collapses the particular correspondence functions. So, the criterion obtained via the convolution of the particular correspondence functions will be called a probabilistic criterion for correspondence (PCC criterion).

We have obtained an analytical dependence for calculating the PCC criterion $W_{np}(p, p_g)$, which differs significantly from the widely used Formula (1). However, it is easy to show that dependence (1) is a special case of dependence (9) if we accept the correspondence functions in the form of linear dependencies (the simplest correspondence functions) shown in Figure 3.

In this case, $s_p(p) = p$ and $s_g(p_g) = p_g$ and, therefore, the *t*-norm will have the form that completely coincides with dependence (1):

$$T(s_p, s_g) = p \cdot p_g$$

From (8), it also follows that the equation of the level line $W_{np}(p, p_g) = const$ is as follows:

$$W_{np}(p, p_g) = \begin{cases} \frac{\ln(1-p)}{\ln(1-W_s)} \exp\left\{-\left|\frac{p_g-1}{\delta}\right|^n\right\}, & if\ p < W_s, \\ \exp\left\{-\left|\frac{p_g-1}{\delta}\right|^n\right\}, & if\ W_s \le p \le 1 \end{cases} = const. \tag{11}$$

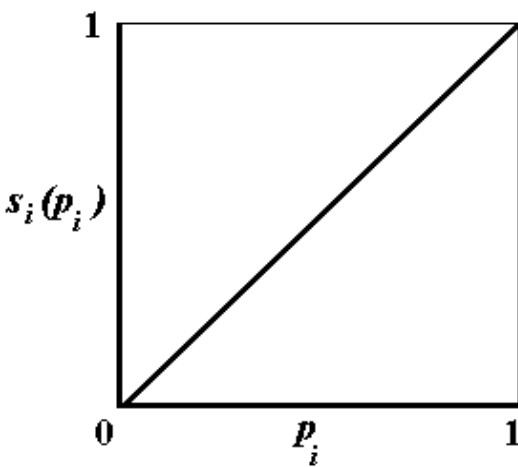

**Figure 3.** The simplest matching functions.

It follows from (10) that when the condition $p \leq W_s$ is fulfilled, the relation between the values of a particular criterion on the same level line can be represented as follows:

$$p = \exp\left[const\cdot\ln(1 - W_s)\exp\left\{\left|\frac{p_g - 1}{\delta}\right|^n\right\}\right].$$

Taking $W_s = 0.95$ as the threshold value, we obtain the following expression for the level line:

$$p = 1 - \exp\left[-3\cdot const\cdot\exp\left\{\left|\frac{p_g - 1}{\delta}\right|^n\right\}\right].$$

Figure 4 shows the initial dependencies $s_p(p)$ and $s_g(p_g)$, as well as the level lines for several values of the PCC criterion (for $n = 2$ and $\delta = 0.25$), obtained by convolving the partial criterion over the *t*-norm.

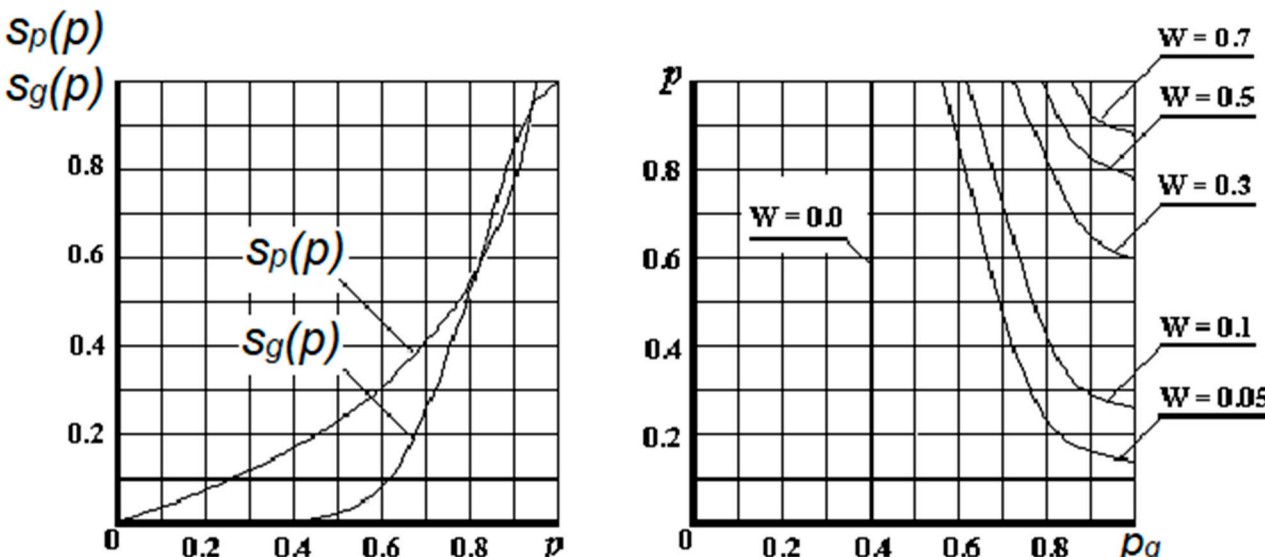

**Figure 4.** Original fit functions and level lines for PCC criterion.

Table 1 shows the coordinates of the PCC criterion level lines, and the columns contain three values of the $W = p\cdot p_g$, criterion, calculated according to dependence (1).

**Table 1.** Coordinates of PCC level lines for different values of *const*.

| Coordinates of the Lines of the Reduced Criterion Level | | | | | | | | | | | |
|---|---|---|---|---|---|---|---|---|---|---|---|
| *Const* = 0.1 | | | *Const* = 0.3 | | | *Const* = 0.5 | | | *Const* = 0.7 | | |
| $p$ 1 | $p_g$ 2 | $p \cdot p_g$ 3 | $p$ 1 | $p_g$ 2 | $p \cdot p_g$ 3 | $p$ 1 | $p_g$ 2 | $p \cdot p_g$ 3 | $p$ 1 | $p_g$ 2 | $p \cdot p_g$ 3 |
| 0.95 | 0.620 | 0.589 | 0.95 | 0.725 | 0.689 | 0.95 | 0.791 | 0.752 | 0.95 | 0.850 | 0.808 |
| 0.90 | 0.643 | 0.578 | 0.90 | 0.757 | 0.681 | 0.90 | 0.836 | 0.752 | 0.90 | 0.924 | 0.831 |
| 0.85 | 0.660 | 0.561 | 0.85 | 0.784 | 0.666 | 0.85 | 0.878 | 0.747 | 0.877 | 1.000 | 0.877 |
| 0.80 | 0.676 | 0.540 | 0.80 | 0.809 | 0.647 | 0.80 | 0.933 | 0.746 | | | |
| 0.75 | 0.690 | 0.518 | 0.75 | 0.835 | 0.626 | 0.777 | 1.000 | 0.777 | | | |
| 0.70 | 0.705 | 0.493 | 0.70 | 0.865 | 0.605 | | | | | | |
| 0.65 | 0.720 | 0.468 | 0.65 | 0.901 | 0.586 | | | | | | |
| 0.60 | 0.735 | 0.441 | 0.60 | 0.966 | 0.580 | | | | | | |
| 0.55 | 0.752 | 0.414 | 0.593 | 1.000 | 0.593 | | | | | | |
| 0.50 | 0.771 | 0.385 | | | | | | | | | |
| 0.45 | 0.7924 | 0.356 | | | | | | | | | |
| 0.4 | 0.8176 | 0.327 | | | | | | | | | |
| 0.35 | 0.8496 | 0.297 | | | | | | | | | |
| 0.3 | 0.8960 | 0.268 | | | | | | | | | |
| 0.259 | 1.00 | 0.259 | | | | | | | | | |

As follows from the analysis of the data given in Table 1, the value of the criterion $W = p \cdot p_g$ along the lines of the level of the PCC criterion changes significantly.

Figure 5 shows the level lines for the PCC criterion (solid line) and the criterion determined by dependence (1) (dashed line). From the analysis of the graph, it follows that for the traditional model obtained via the convolution of the partial criterion in the form of $W(p, p_g) = p \cdot p_g$ according to formula (3), options 1 and 2 are equivalent. At the same time, according to the PCC criterion, these variants differ significantly.

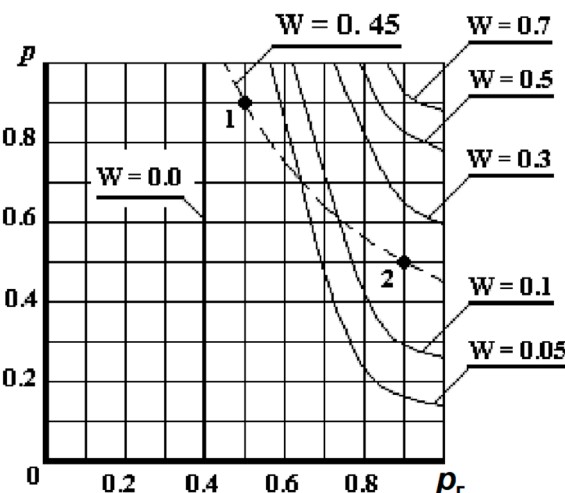

**Figure 5.** Level lines for criteria.

As an example, Figure 6 shows the surface of the PCC criterion $W_{np}(p, p_g)$, built according to the proposed method.

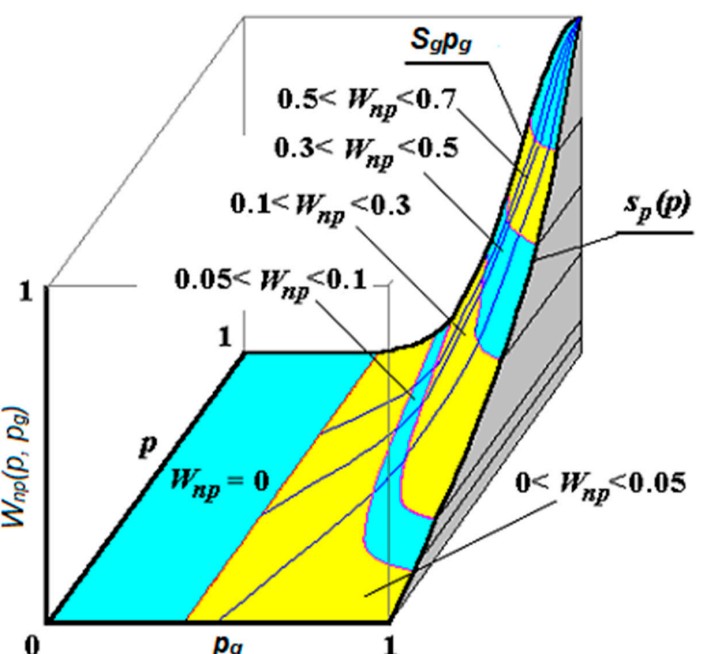

**Figure 6.** Criterion surface with level lines when convolving partial indicators using PCC criterion.

## 5. Discussion

Hamacher *t*-norms [26,27], which depend on the parameter $\gamma(0 \leq \gamma < \infty)$, are also interesting:

$$W(p, p_g) = \frac{s_p(p) \cdot s_p(p_g)}{\gamma + (1 - \gamma) \cdot \left[s_p(p) + s_p(p_g) - s_p(p) \cdot s_p(p_g)\right]}. \tag{12}$$

As it follows from (11), when the value of the parameter $\gamma = 1$, this *t*-norm turns into the *t*-norm considered above, i.e., into the convolution of two correspondence functions $W(p, p_g) = s_p(p) \times s_p(p_g)$, and in the case of $\gamma = 0$ turns into the *t*-norm of the following form:

$$W(p, p_g) = \frac{s_p(p) \cdot s_p(p_g)}{s_p(p) + s_p(p_g) - s_p(p) \cdot s_p(p_g)}. \tag{13}$$

The free parameter $\gamma$ allows for changing the shape of the level lines and thereby obtaining solutions that are more adequate to the processes under study.

As was noted earlier, this article was devoted to assessing the effectiveness of managing one specific type of UAV. This is due to the fact that they are widely used in traditional areas such as logistics, monitoring of the environment and industrial facilities, search and rescue operations, etc. [28]. In addition, recently, UAVs have been being used in completely new areas: monitoring crop diseases, landslides, mountain ranges, etc. [29–31]. The approach, with minimal changes, can be adapted to assess the effectiveness of UAVs in these areas as well. The main difficulty in the practical application of this approach is the need to collect statistical information on the reliability of various types of UAVs and the probabilities of their performance related to functional tasks in all stages of flight. However, this task is now being solved on the basis of the expert assessment of specialists and available information from UAV manufacturers. Another development of the proposed approach is its adaptation in relation to a swarm of UAVs; this task is also at the stage of completion and will be issued in the form of a separate article in the near future.

## 6. Conclusions

The proposed method for evaluating the effectiveness of a UAV flight control system has the following advantages:

1.  Instead of the private criterion $p_i$, the correspondence functions $s_i(p_i)$ are used, which adequately reflect the significance of the advising private criterion;
2.  The PCC criterion is free from the shortcomings of the convolution criterion of the form of (1), but retains its structure and has a clear physical meaning; the convolution is carried out for the correspondence functions of a particular probabilistic criterion;
3.  It is not necessary to introduce strict restrictions on the $p_i$, which, when using criterion (1), are usually formulated as $p_i \geq p_i^*$ and can lead to the loss of a rational solution when $p_i$ is in the region close to the threshold value;
4.  The technique is valid not only for the convolution of two correspondence functions, but also in accordance with expression (7) for an arbitrary number of them due to the associativity property of the $t$-norm;
5.  The methodology is quite universal and, with minimal changes, can be adapted to assess the effectiveness of various CTSs.

**Author Contributions:** Conceptualization, V.V., M.K. and V.K.; methodology, V.V., M.K. and V.K.; software, V.V., M.K. and V.K.; validation, V.V., M.K. and V.K.; formal analysis, V.V., M.K. and V.K.; investigation, V.V., M.K. and V.K.; resources, V.V., M.K. and V.K.; data curation, V.V., M.K. and V.K.; writing—original draft preparation, V.V., M.K. and V.K.; writing—review and editing, V.V., M.K. and V.K.; visualization, V.V., M.K. and V.K.; supervision, V.V., M.K. and V.K.; project administration, V.V., M.K. and V.K.; funding acquisition, V.V., M.K. and V.K. All authors have read and agreed to the published version of the manuscript.

**Funding:** This research was funded by the Russian Science Foundation, grant number 23-69-10084, https://rscf.ru/project/23-69-10084/, accessed 20 July 2023.

**Data Availability Statement:** Not applicable.

**Conflicts of Interest:** The authors declare no conflict of interest.

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
