# Peer review of "Evaluation of Unmanned-Aerial-Vehicle-Integrated Control System Efficiency on the Basis of Generalized Multiplicative Criterion"

_inventions, doi:10.3390/inventions8040094_

Round 1

Reviewer 1 Report

The authors have proposed an approach to evaluating the effectiveness of UAV flight control system using a generalized multiplicative criterion. The proposed method is well-described in the paper. The authors have also discussed the theoretical basis of the proposed method. The relevant research works are reviewed in the paper. To sum up, the paper is well-structured and presents a method that can be used to evaluate the effectiveness of UAV flight control system.

Please address the following issues:
1. The quality of the figures should be improved - all of them are
   pixelated.
2. Please explain in more details the possibility of using the proposed method in other areas, which was mentioned briefly at the end of section 1.
3. Please analyze the weak points of the proposed approach.

The English language should be improved when it comes to grammar and style.

Author Response

1. The quality of the figures should be improved - all of them are pixelated.

The article was written under the grant of the Russian Science Foundation No. 23-69-10084, one of the requirements of which is the use of Russian software. Therefore, all calculations were carried out using SimInTech (Russian analogue of MatLab), which is still being finalized. In this regard, the quality of some figures that are formed directly in the course of calculations may not have high quality enough.

2. Please explain in more details the possibility of using the proposed method in other areas, which was mentioned briefly at the end of section 1.

This article was written under the grant of the Russian Science Foundation No. 23-69-10084 for the development of data management models in heterogeneous networks, and the proposed approach is applicable to almost any mobile subscriber, including ground and surface unmanned vehicles. Relevant explanations are included in the Introduction section.

3. Please analyze the weak points of the proposed approach.

We made changes in the Discussion section.

Reviewer 2 Report

Reviewer’s Report on the manuscript entitled:

Evaluation of UAV Integrated Control System Efficiency on the Basis of Generalized Multiplicative Criterion

The authors proposed a method to evaluate the effectiveness of UAV system based on a generalized multiplicative criterion.  While the topic and results are interesting, the presentation and structure of the manuscript are poor and should be improved. The Introduction and literature review also need improvement. Please see below my comments.

Poor grammar. Please rewrite lines 9-11. For example, “This article proposes an approach to evaluate the effectiveness of UAV system…”

In Introduction, please describe machine learning methods, UAV, and their application for crop disease monitoring and landslide monitoring:

https://doi.org/10.3390/rs14051239

https://doi.org/10.3390/s23094287

https://doi.org/10.4408/IJEGE.2019-01.S-12

Line 25 and 26. I suggest expanding this paragraph and discussing more about UAV and its applications. Please include the following most recent articles

Lines 25-30. Please define UAV, UOM, UTM, etc. All the acronyms must be defined the first time they appear in the manuscript.

Line 89. Please describe how the rest of the manuscript is organized.

Lines 131, 135. Do you mean (i = 1,…,n)? ‘Please correct.

Lines 165, 227, 254. Style issue. Please replace “[0.1]” with “[0, 1]”

Line 292. Please remove “The” before Figure. Please check and correct such issues elsewhere.

Line 318. Please remove one of the equality signs: change “= =” to “=”

Please add a flowchart for your method in the method section.

Please elaborate on your results in the light of other similar studies in the Discussion section. Please also include the limitations of your work and give future direction.

Please carefully proofread the manuscript

Thank you

There are many grammar/punctuation issues that should be carefully checked and corrected.

Author Response

1. The authors proposed a method to evaluate the effectiveness of UAV system based on a generalized multiplicative criterion.While the topic and results are interesting, the presentation and structure of the manuscript are poor and should be improved. The Introduction and literature review also need improvement. 

We made changes in the Introduction and Discussion sections. The list of references has been also expanded.

2. Poor grammar. Please rewrite lines 9-11. For example, “This article proposes an approach to evaluate the effectiveness of UAV system…”

The article has been revised. The above-mentioned sentence has been rewritten.

3. In Introduction, please describe machine learning methods, UAV, and their application for crop disease monitoring and landslide monitoring.

This article does not consider machine learning methods, according to the areas of application of UAV we made changes in the Discussion and Literature review sections. 

4. Line 25 and 26. I suggest expanding this paragraph and discussing more about UAV and its applications. Please include the following most recent articles:

https://doi.org/10.3390/rs14051239

https://doi.org/10.3390/s23094287

https://doi.org/10.4408/IJEGE.2019-01.S-12

We made changes in the Discussion section, information about UAV has been added there, these and other articles have been added to the Literature section.

5. Lines 25-30. Please define UAV, UOM, UTM, etc. All the acronyms must be defined the first time they appear in the manuscript.

The definition of acronyms is added in the first appearance in the text.

6. Line 89. Please describe how the rest of the manuscript is organized. 

We made changes in the Introduction section.

7. Lines 131, 135. Do you mean (i = 1,…,n)? ‘Please correct.

The error is corrected.

8. Lines 165, 227, 254. Style issue. Please replace “[0.1]” with “[0, 1]”

The error is corrected.

9. Line 292. Please remove “The” before Figure. Please check and correct such issues elsewhere. 

The errors are corrected.

10. Line 318. Please remove one of the equality signs: change “= =” to “=”

The error is corrected.

11. Please add a flowchart for your method in the method section.

In the Methodology section, after the second paragraph, the sequence of solving the problem of determining the global efficiency criterion has been added.

12. Please elaborate on your results in the light of other similar studies in the Discussion section.

We made changes in the Discussion section.

13. Please also include the limitations of your work and give future direction.

We made changes in the Discussion section.

Reviewer 3 Report

The paper discusses some issues concerning the control systems for unmanned aerial vehicles. The paper requires some improvements and extensive proofreading. I can point out the following issues:

1. The definition of "CTS efficiency" is weird and vague. Please, clarify the term "efficiency" in your study and clearly define the object and subject of your research in combination with targets and goals.

2. All abbreviations and acronyms should be defined prior to their appearance in the text.

3. Can the proposed technique be applied to the assessment of surface unmanned vehicles (robotic boats) performance? For example, for estimating energy efficiency and autonomy range of the battery-powered small unmanned surface vehicles

4. How the "economic feasibility" can be assessed for a certain UAV? Please, provide some quantitative metrics.

5. In my opinion, the provided math can be applied to almost arbitrary mobile agent or technical object. What is the UAV specifics here?

6. What is the "clear physical meaning" of "global criterion 1"? Please, clarify this point further.

7. Please, clarify the term "a real one-place function" and where is the proof of its "continuousity". 

8. Methodology (section 3) should be clarified. The statements are strong but weakly grounded.

9. What is the purpose of Fig.3? To show the linear dependence to the reader?

10. Experimental confirmation of the proposed approach is missing. Did the Authors test this technique on real UAVs?

Taking all the abovementioned into account, I believe the paper requires a lot of revisions prior to publication.

English style and language are far from desirable standards. I recommend extensive proofreading of the manuscript.

Author Response

1. The definition of "CTS efficiency" is weird and vague. Please, clarify the term "efficiency" in your study and clearly define the object and subject of your research in combination with targets and goals.

The authors did not give a new definition of the term "CTS efficiency", but used generally accepted formulations with reference to sources. At the end of the third paragraph of the Introduction section, a clarification was made regarding the effectiveness of CTS.

2. All abbreviations and acronyms should be defined prior to their appearance in the text.

The definition of acronyms is added in the first appearance in the text.

3. Can the proposed technique be applied to the assessment of surface unmanned vehicles (robotic boats) performance? For example, for estimating energy efficiency and autonomy range of the battery-powered small unmanned surface vehicles.

This article was written under the grant of the Russian Science Foundation No. 23-69-10084 for the development of data management models in heterogeneous networks, and the proposed approach is applicable to almost any mobile subscriber, including ground and surface unmanned vehicles. Relevant explanations are included in the Introduction section.

4. How the "economic feasibility" can be assessed for a certain UAV? Please, provide some quantitative metrics.

In the third paragraph of the Introduction section, there is the explanation that the issues of assessing economic feasibility are not considered in this article, but only feasibility in the aspect of fulfilling those targets for which it is being developed.

5. In my opinion, the provided math can be applied to almost arbitrary mobile agent or technical object. What is the UAV specifics here?

This article was written under the grant of the Russian Science Foundation No. 23-69-10084 for the development of data management models in heterogeneous networks and the proposed approach is applicable to almost any mobile subscriber. The article considers the application of the developed methodology in relation to one of the possible types of mobile subscribers: UAV. We made changes in the Introduction section.

6. What is the "clear physical meaning" of "global criterion 1"? Please, clarify this point further.

We made changes: the global criterion is the product of particular criteria.

7. Please, clarify the term "a real one-place function" and where is the proof of its "continuousity". 

An explanation has been added to the text in the Theoretical basis section: a one-place function is a function of one argument. Continuity as applied to the fit and reliability functions are illustrated in Figures 1 and 2.

8. Methodology (section 3) should be clarified. The statements are strong but weakly grounded.

We made changes in the Methodology section.

9. What is the purpose of Fig.3? To show the linear dependence to the reader?

It is just as an illustration of the simplest match function.

10. Experimental confirmation of the proposed approach is missing. Did the Authors test this technique on real UAVs?

At this stage, only theoretical studies and the collection of statistical data are carried out, including on the reliability of various types of UAV.

Round 2

Reviewer 2 Report

I thank the authors for addressing my comments and improving their manuscript. I think the manuscript can be accepted after some editorial corrections. For example:

Line 12. Grammar issue. "criteria" not "criterion". Similarly for line 200. Note that "criteria" is plural and "criterion" is singular.

Line 346. Please remove the phrase " The authors understand that". Simply say "The main difficulty..."

Thank you!

Please carefully proofread the manuscript as there are some grammar/typos/punctuation issues.

Author Response

Dear reviewer,

Thank you very much for your comments. We corrected the paper following your recommendations.

Best regards,

Authors

Reviewer 3 Report

Dear Authors,

Thank you very much for providing a revised version of your manuscript.

I am generally satisfied with the revisions and point-by-point reply letter. However, the language and style of the paper require a bit of polishing.

I wish the Authors good luck in their further studies.

Sincerely,

Reviewer

The language and style of the paper are to be improved further.

Author Response

(The authors gave the same response as above.)
